# DNA damage induces nuclear actin filament assembly by Formin-2 and Spire-1/2 that promotes efficient DNA repair

Brittany J Belin[1,2], Terri Lee[1], R Dyche Mullins[1,2]*

[1]Department of Cellular and Molecular Pharmacology, University of California, San Francisco, San Francisco, United States; [2]Physiology Course, Marine Biological Laboratory, Woods Hole, United States

**Abstract** Actin filaments assemble inside the nucleus in response to multiple cellular perturbations, including heat shock, protein misfolding, integrin engagement, and serum stimulation. We find that DNA damage also generates nuclear actin filaments—detectable by phalloidin and live-cell actin probes—with three characteristic morphologies: (i) long, nucleoplasmic filaments; (ii) short, nucleolus-associated filaments; and (iii) dense, nucleoplasmic clusters. This DNA damage-induced nuclear actin assembly requires two biologically and physically linked nucleation factors: Formin-2 and Spire-1/Spire-2. Formin-2 accumulates in the nucleus after DNA damage, and depletion of either Formin-2 or actin's nuclear import factor, importin-9, increases the number of DNA double-strand breaks (DSBs), linking nuclear actin filaments to efficient DSB clearance. Nuclear actin filaments are also required for nuclear oxidation induced by acute genotoxic stress. Our results reveal a previously unknown role for nuclear actin filaments in DNA repair and identify the molecular mechanisms creating these nuclear filaments.

*For correspondence: Dyche.Mullins@ucsf.edu

**Competing interests:** The authors declare that no competing interests exist.

## Introduction

Actin was first identified in muscle over seventy years ago (*Straub, 1942*) and has been established as a component of non-muscle cells for nearly half a century (*Hatano and Oosawa, 1966*). Subsequent work revealed how actin filaments help organize the cytoplasm of all eukaryotic cells, supporting many fundamental biological processes, including: motility, division, phagocytosis, endocytosis, and membrane trafficking. The first reports of actin inside the nucleus of a cell appeared forty years ago (*LeStourgeon et al., 1975*), and since that time, actin has been found in the nuclei of many different cell types, linked to a variety of nuclear processes (*Pederson and Aebi, 2002*).

Recent work has identified the molecular mechanisms that control the nuclear concentration of actin, uncovering new roles for the actin-binding proteins profilin and cofilin as co-factors for actin's nucleocytoplasmic transport. The nuclear export factor exportin-6 (XPO6) binds profilin-actin complexes in the nucleus—as well as a handful of other actin-binding proteins—and shuttles them into the cytoplasm (*Stüven et al., 2003; Bohnsack et al., 2006*). Conversely, nuclear import of actin is regulated by importin-9 (IPO9), which transports cofilin–actin complexes from the cytoplasm into the nucleus (*Dopie et al., 2012*).

In many organisms, the export factor XPO6 is not expressed in the oocytes of, and so the germinal vesicles contain a high concentration of actin (*Stüven et al., 2003*). In *Xenopus laevis* oocytes, this germinal vesicle actin forms a filamentous mesh that protects nucleoli from gravity-induced aggregation (*Feric and Brangwynne, 2013*). Actin filaments associated with germinal vesicles of starfish oocytes facilitate nuclear envelope breakdown and form a contractile net that facilitates chromosome capture during mitosis (*Lénárt et al., 2005; Mori et al., 2014*). Several studies have also implicated nuclear actin filaments in oocyte transcription (reviewed in *Belin and Mullins, 2013*).

**eLife digest** In animals, plants, and other eukaryotic organisms, a cell's DNA is contained within a structure called the nucleus, which separates it from the rest of the interior of the cell. Filaments of a protein called actin are normally found outside the nucleus, where they help give the cell its overall shape and organize its contents. However, these filaments can sometimes form inside the nucleus in response to a sudden increase in heat or another type of stress. However, it was not clear what role these actin filaments play in the nucleus because it was difficult to distinguish them from the actin filaments that form in other parts of the cell.

Researchers have recently developed new techniques to study actin filaments inside the nuclei of live cells under a microscope, using fluorescent protein tags. Here, Belin et al.—including some of the researchers involved in the previous work—used this technique to investigate whether DNA damage causes actin filaments to form in the nuclei of human cells.

The experiments show that DNA damage does indeed lead to the formation of actin filaments in the nucleus. In a structure within the nucleus called the nucleolus, the actin filaments are short. However, in the rest of the nucleus, the actin forms long filaments and dense clusters. Cells that contained lower levels of actin were less able to repair their DNA than normal cells.

Belin et al. also identified three proteins—called Formin-2, Spire-1, and Spire-2—that assemble the actin filaments in the nucleus. These proteins are also required to make actin filaments in other parts of the cell. The experiments show that the level of Formin-2 increases in the nucleus after DNA damage, and that the DNA of cells lacking this protein is more severely damaged. Belin et al.'s findings reveal a new role for actin in the repair of DNA and the next challenge is to understand the details of how this works.

In contrast, most somatic cells express some amount of XPO6, and therefore, have a much lower concentration of actin in the nucleus than in the cytoplasm. Also, unlike *Xenopus* germinal vesicles, mammalian somatic nuclei contain relatively small amounts of filamentous actin (*Belin et al., 2013*), suggesting that monomeric actin may play an important role. Monomers of actin and several actin-related proteins (Arps), for example, are conserved components of chromatin-remodeling complexes (*Farrants, 2008*), and nuclear actin monomers inhibit the activity of the serum-responsive transcriptional co-activator MRTF (myotonin-related transcription factor) (*Vartiainen et al., 2007*; *Mouilleron et al., 2008*). Many reports have also linked actin to the regulation of RNA polymerases, although there are conflicting data on whether this activity depends on monomers or filaments (*Belin and Mullins, 2013*).

Functions for filamentous actin in somatic cell nuclei are slowly beginning to emerge. Serum stimulation of quiescent fibroblasts (*Baarlink et al., 2013*) and integrin engagement in spreading cells (*Plessner et al., 2015*) induce transient (<60 s) bursts of nuclear actin polymerization, driven by the nucleation activity of formin-family proteins mDia1 and mDia2. These short-lived filaments appear to promote activity of the transcriptional co-activator MRTF by depleting monomeric actin from the nucleus. Serum stimulation also activates the actin-severing protein MICAL-2, which reversibly oxidizes actin monomers, rendering them incapable of inhibiting MRTF-dependent transcription (*Lundquist et al., 2014*). Environmental stresses also promote actin assembly in somatic cell nuclei. Heat shock, dimethyl sulfoxide (DMSO), depletion of ATP, and oxidative stress all induce formation of nuclear filament bundles that contain large amounts of cofilin (*Fukui, 1978*, *Fukui and Katsumaru, 1980*; *Iida et al., 1992*; *Pendleton et al., 2003*; *Kim et al., 2009*). In addition to its function as a co-factor for nuclear import, cofilin appears to play a structural role in these cofilin–actin rods, which are highly oxidized and appear to be held together by intermolecular disulfide bonds between cofilin molecules (*Pfannstiel et al., 2001*; *Bernstein et al., 2012*; *Zhang et al., 2013*). Little is known about the physiological role of these cofilin–actin rods but they sense and perhaps regulate the reducing potential of the nucleus (*Bernstein et al., 2012*; *Munsie et al., 2012*).

Many functions proposed for nuclear actin have been controversial, due in part to a lack of molecular tools for visualizing and perturbing actin inside the nucleus without affecting cytoplasmic actin (*Belin et al., 2013*). The discovery of actin's nuclear import and export factors, along with the recent identification of some of the molecular mechanisms that create nuclear actin filaments, now

enable us to make more specific perturbations of actin inside the nucleus. In addition, we and others have developed fluorescent probes that enable us to visualize actin monomers and filaments in the nuclei of live cells (*Baarlink et al., 2013*; *Belin et al., 2013*; *Plessner et al., 2015*).

Using these recently developed tools, we discovered that DNA damage induced by various genotoxic agents triggers formation of actin filaments inside the nucleus of mammalian cells. These filaments promote efficient repair of DNA double-strand breaks (DSBs) and are required for a DNA damage-associated burst of oxidation in the nucleus. DNA damage-induced nuclear actin structures differ in both composition and mechanism of assembly from those triggered by serum stimulation or by non-specific cell stresses. Specifically, we find that the actin regulators Formin-2 (FMN2) and Spire-1/2 nucleate nuclear actin assembly in response to DNA damage. Homologs of Formin-2 and Spire-1/2 interact directly (*Quinlan et al., 2007*; *Vizcarra et al., 2011*; *Montaville et al., 2014*) and collaborate to form functional actin networks in mouse and *Drosophila* oocytes. Murine oocyte FMN2 and Spire are required for migration of meiotic spindles to the cortex, extrusion of polar bodies, and radial transport of vesicles (*Schuh and Ellenberg, 2008*; *Schuh, 2011*). In *Drosophila*, homologs of FMN2 (Cappuccino) and Spire collaborate to build actin networks required for oocyte polarity (*Quinlan et al., 2005*; *Bor et al., 2015*). We suggest that DNA damage-induced nuclear actin filaments may facilitate movement of chromatin and repair factors after DNA damage.

## Results

### DNA damage induces formation of nuclear actin filaments

Previously, we developed and validated a fluorescent probe, Utr230-EGFP-3×NLS (Utr230-EN), capable of imaging actin filaments inside nuclei of live cells (*Belin et al., 2013*). In mammalian tissue culture cells, the distribution of Utr230-EN exhibits one of three basic patterns: (i) accumulation on cytoplasmic stress fibers and diffuse localization throughout the nucleoplasm (~53% of cells); (ii) cytoplasmic stress fibers and punctate structures in the nucleoplasm (~40% of cells); and (iii) primarily cytoplasmic localization to stress fibers and/or aggregates at the nuclear periphery (~7% of cells) (*Figure 1—figure supplement 1*). Further analysis revealed that the punctate nucleoplasmic structures recognized by Utr230-EN are endogenous structures that contain short (<0.5 μm) actin filaments (*Belin et al., 2013*) (*Figure 1A*). To characterize these nuclear actin filaments, we performed an immunofluorescence screen to test for co-localization with various nuclear landmarks; among the co-localization candidates, we selected was Rap1, a marker for telomeres. We did not detect significant co-localization of nuclear actin with intact telomeres in HeLa cells, but when we uncapped the telomeres using MT-hTer-47A RNAi (*Marusic et al., 1997*; *Li et al., 2004*), we reproducibly observed the formation of larger nuclear actin structures, including the formation of long (>1 μm) filaments (*Figure 1B*).

Uncapped telomeres resemble double-strand DNA breaks, so we hypothesized that nuclear actin assembly might be triggered as part of a more general DNA damage response. To test this idea, we damaged HeLa cell DNA using a variety of genotoxic agents: incubation with 0.01% methyl methanesulfonate (MMS); incubation with 50 pg/ml neocarzinostatin (NCS); or exposure to 50 J/m-s ultraviolet radiation. These various treatments all triggered assembly of actin filaments in the nucleus (*Figure 1B*). We compared nuclear actin filaments generated by telomere uncapping with those induced by genotoxic agents by counting the fraction of cells that contain nuclear filaments longer than 1 μm as well as the mean length of these long nuclear filaments in cells treated with either MT-hTer-47A for 5–7 days or 0.01% MMS for 2 hr. The fraction of cells with long filaments; the mean filament length; and the standard deviation of filament length were similar after the two treatments (*Figure 1C–G*). Further analysis of nuclear filaments induced by MMS revealed the formation of three distinct classes of nuclear actin structure: (i) elongated nucleoplasmic filaments (>1 μm); (ii) amorphous, nucleoplasmic clusters; and (iii) nucleolus-associated filaments (*Figure 1H*). The third class of structures includes both peri- and intra-nucleolar filaments, as judged by co-localization with the nucleolus marker, fibrillarin (*Figure 1I*).

To verify that DNA damage-induced nuclear actin filaments are endogenous structures, and not artifacts produced by expression of the Utr230-EN probe, we visualized them using other probes for filamentous actin. In HeLa cells not expressing Utr230-EN, fluorescent derivatives of phalloidin faintly but reproducibly detected all three classes of DNA damage-induced nuclear filaments (*Figure 2A*). Detecting fluorescent phalloidin in the nucleus following DNA damage, however, presented technical challenges. High concentrations of fluorescent phalloidin bound to actin filaments in the cytoplasm

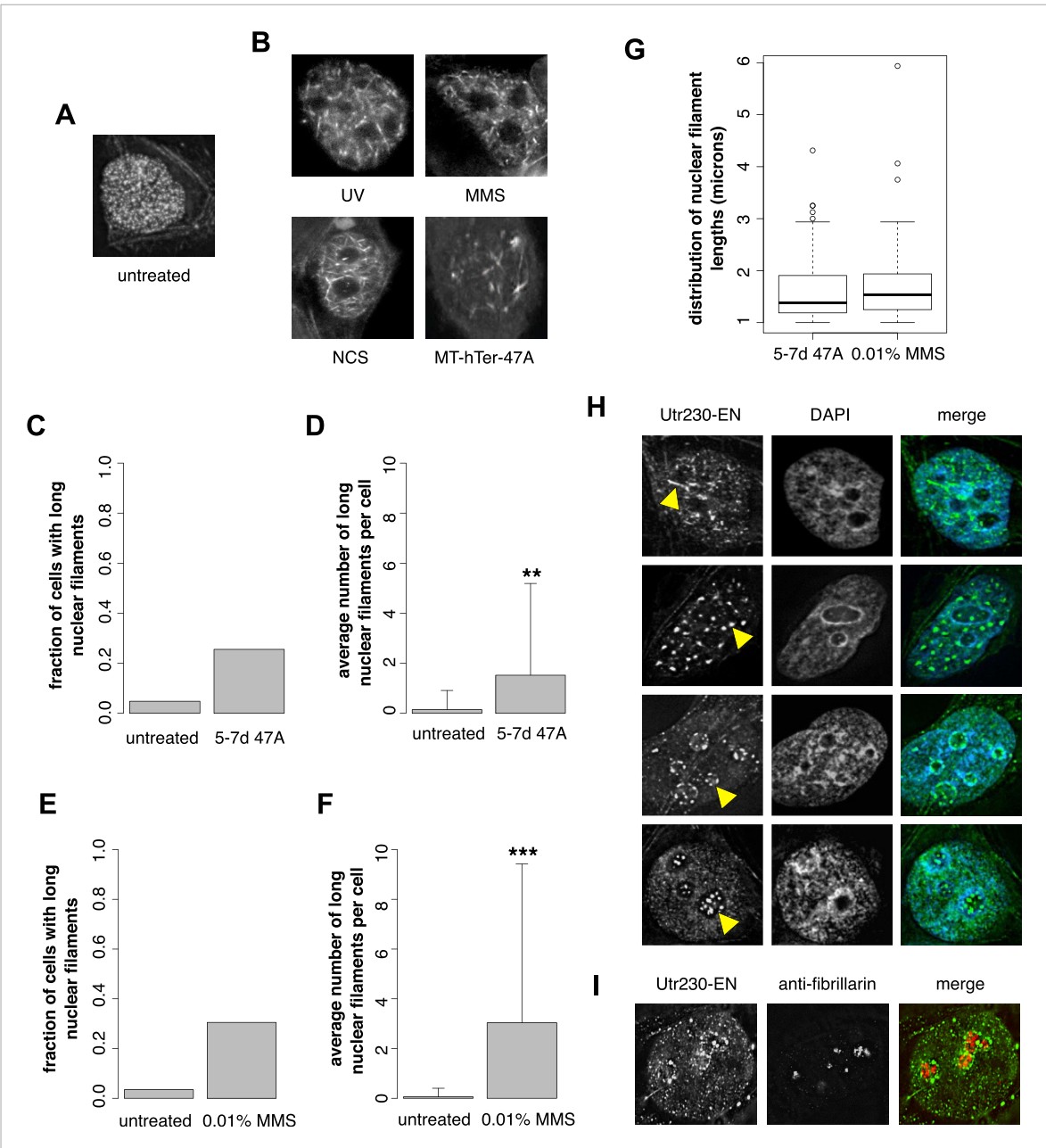

**Figure 1**. DNA damage induces nuclear actin polymerization. (**A**) Localization of Utr230-EGFP-NLS (Utr230-EN) in unstressed HeLa cells. (**B**) Utr230-EN localization after induction of DNA damage via: 50 J/m-s UV exposure, 120-min incubation in media supplemented with 0.01% methyl methanosulfonate (MMS), 120-min incubation in media supplemented with 50 pg/ml neocarzinostatin (NCS), or 1 week post-transfection with telomere uncapping reagent MT-hTer-47A. (**C**) Percentage of cells with long (>1 micron) nuclear filaments after treatment with telomere uncapping reagent MT-hTer-47A (47A) for 5–7 days. N = 83–90 cells. (**D**) Average number of long nuclear filaments per cell after treatment with telomere uncapping reagent 47A for 5–7 days. N = 83–90 cells. (**E**) Percentage of cells with long (>1 micron) nuclear filaments after incubation in 0.01% MMS for 120′. N = 87–118 cells. (**F**) Average number of long nuclear filaments per cell incubation in 0.01% MMS for 120′. N = 87–118 cells. (**G**) Comparison of length distributions of long nuclear filaments after treatment with either 47A for 5–7 days or 120′ in 0.01% MMS. Open circles indicate outliers. N = 94–95 filaments. (**H**) Utr230-EN localization after 120 minute incubation in 0.01% MMS including several classes of nuclear filaments, listed from top to bottom: elongated nucleoplasmic, clustered, peri- and intra-nucleolar. Nuclear filaments are indicated with yellow arrows. (**I**) Co-localization of peri-nucleolar filaments with an antibody detecting nucleolar protein fibrillarin. Asterisks indicate p-values < 10E-3 (**) or 10E-4 (***) for all panels.

The following figure supplement is available for figure 1:

**Figure supplement 1**. Distribution of Utr230-EN localization patterns in HeLa cells.

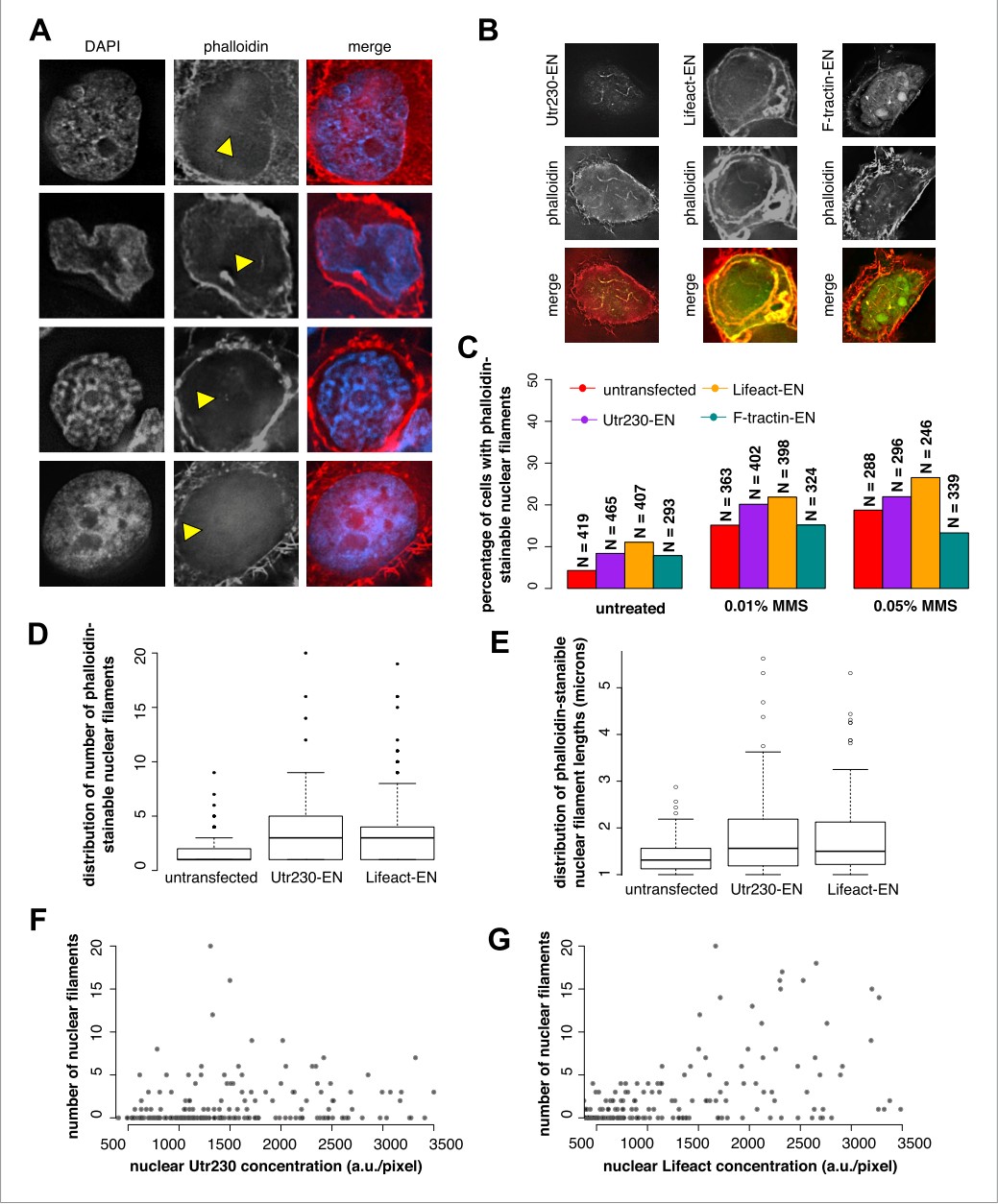

**Figure 2**. Comparison of nuclear-targeted actin reporters for detection of DNA damage-induced nuclear actin filaments. (**A**) Faint detection of elongated nuclear actin filaments in the nucleoplasm and peri- and intra-nucleolar regions after 0.01% MMS treatment by phalloidin in untransfected cells. Arrows indicate locations of nuclear filaments, including elongated nucleoplasmic filaments (top two rows) and phalloidin staining of intra-nucleolar (third row) and peri-nucleolar (bottom row) structures. (**B**) Detection of elongated nuclear actin filaments after 0.01% MMS treatment by nuclear-targeted variants of three common live-cell actin reporters: Utr230-EN, Lifeact-EGFP-NLS (Lifeact-EN), and F-tractin-EGFP-NLS (F-tractin-EN). (**C**) Percentage of cells containing phalloidin-stainable nuclear filaments before and after MMS treatment within cell lines expressing actin reporters shown in (**B**). N values are as shown. (**D**) Distribution of the number of phalloidin-stainable nuclear filaments per cell after MMS treatment within cell lines expressing actin reporters shown in (**B**). Closed circles indicate outliers. (**E**) Distributions of lengths of long (>1 micron) phalloidin-stainable nuclear filaments per cell after MMS treatment within cell lines expressing actin reporters shown in (**B**). N = 82–100 filaments. Open circles indicate outliers. (**F**) Scatter plot depicting the number of Utr230-EN detected nuclear filaments per cell as a function of the concentration of Utr230-EN in the nucleus following treatment with 0.01% MMS for 120 minute. N = 206 cells. (**G**) Scatter plot depicting the number of Lifeact-EN detected nuclear filaments per cell as a function of the concentration of Lifeact-EN in the nucleus following treatment with 0.01% MMS for 120′. N = 214 cells.

swamp the much smaller signal from the nucleus. For example, the faint phalloidin fluorescence coming from the nucleus was difficult to detect by confocal microscopy, in part due to the relatively low light throughput of the pinholes. Wide-field deconvolution microscopy provided much more signal and enabled us to more easily observe the faint signal from phalloidin-bound filaments in the nucleus.

We also detected DNA damage-induced nuclear actin filaments using Enhanced Green Fluorescent Protein (EGFP) fusions of the actin-binding peptides Lifeact and F-tractin (*Schell et al., 2001*; *Riedl et al., 2008*) targeted to the nucleus (*Figure 2B*). Using these probes, however, we detected filaments in a smaller fraction of 0.01% MMS-treated cells than in our Utr230-EN cell line. This is likely due to the fact that the higher fluorescence baseline of Lifeact-EN and F-tractin-EN in the nucleus obscures filaments (*Figure 2B*), and this is consistent with a parallel study in which we observed much higher levels of diffuse fluorescence from Lifeact- and F-tractin-based probes in the cytoplasm compared to Utr230 (*Belin et al., 2015*). To test this interpretation, we calculated the percentage of cells with phalloidin-stainable nuclear filaments in untransfected cells and cell lines expressing Utr230-, Lifeact-, or F-tractin-EN. In all cases, ~5% of untreated cells contain phalloidin-stainable nuclear actin, whereas after induction of DNA damage by 0.01% or 0.05% MMS incubation, the percent of cells with phalloidin-stainable nuclear actin increases to ~15% and ~20%, respectively (*Figure 2C*). Thus, the low frequency of cells with nuclear filaments detected by Lifeact- and F-tractin-EN does not correlate with fewer cells containing phalloidin-stainable nuclear actin. These results indicate that the DNA damage-induced nuclear filaments recognized by Utr230-EN are endogenous structures and that expression of nucleus-targeted actin probes does not significantly perturb nuclear actin assembly.

When we limited our analysis to cells that contain large (>1 μm) nuclear actin structures, we observed—by all of our labeling methods—that the average number of actin structures per nucleus also increases significantly after DNA damage. We noticed, however, that the distribution filaments per nucleus is skewed toward both higher filament counts and longer filament lengths in cells expressing Lifeact-EN or Utr230-EN (*Figure 2D,E*). In the Lifeact-EN cell line, high filament density in the nucleus correlates with higher Lifeact-EN expression levels, suggesting that Lifeact stabilizes filaments or enhances polymerization rate (*Figure 2G*). The number of filaments per nucleus, however, does not correlate with Utr230-EN expression level (*Figure 2F*), suggesting that this probe does not promote filament formation in the same way as Lifeact. It is possible that Utr230-EN expression enhances the DNA damage response or that the effect of this probe on nuclear filament stability saturates at very low concentrations, below those required for live-cell imaging. Given the poor contrast of the F-tractin-EN and Lifeact-EN probes and the concentration-dependent stabilizing effects of Lifeact-EN, we regard Utr230-EN as the best available tool for imaging nuclear actin filaments generated by DNA damage.

## Nuclear actin polymerization is required for efficient clearance of double-strand DNA breaks

Many DNA repair proteins have been identified by their effects on the kinetics of DSB detection and clearance. To determine whether nuclear actin plays a role in DNA repair, we compared the number of DSBs induced by MMS in cells with different concentrations of nuclear actin. Initially, we quantified DSBs by expressing a fluorescently tagged fragment of 53BP1, a commonly used live-cell marker for DSBs (*Dimitrova et al., 2008*). In our hands, however, the number of fluorescent foci generated by this 53BP1 reporter correlates directly with the expression level of the construct, making it unsuitable for quantitative studies (*Figure 3—figure supplement 1A*). Instead, we quantified DSBs by immunofluorescence, using antibodies against either 53BP1 or histone variant pS139 H2AX. This method yielded consistent DSB counts across multiple replicate samples treated with the same dose of MMS (*Figure 3A*).

To modulate the concentration of actin in the nucleus, we knocked down expression of either the nuclear import factor, IPO9, or the nuclear export factor, XPO6. For these experiments, we employed siRNAs previously validated in human cells (*Bohnsack et al., 2006*; *Dopie et al., 2012*; *Belin et al., 2013*). Knocking down expression of XPO6 increased the fraction of cells with detectable nuclear actin structures as well as the average number of nuclear structures in each cell. This increase occurred in both untreated cells and cells treated with MMS. Conversely, cells from which IPO9 had been depleted lacked detectable nuclear actin structures, regardless of whether they were treated with MMS (*Figure 3—figure supplement 1B,C*). Loss of XPO6 had no effect on the number of DSB foci

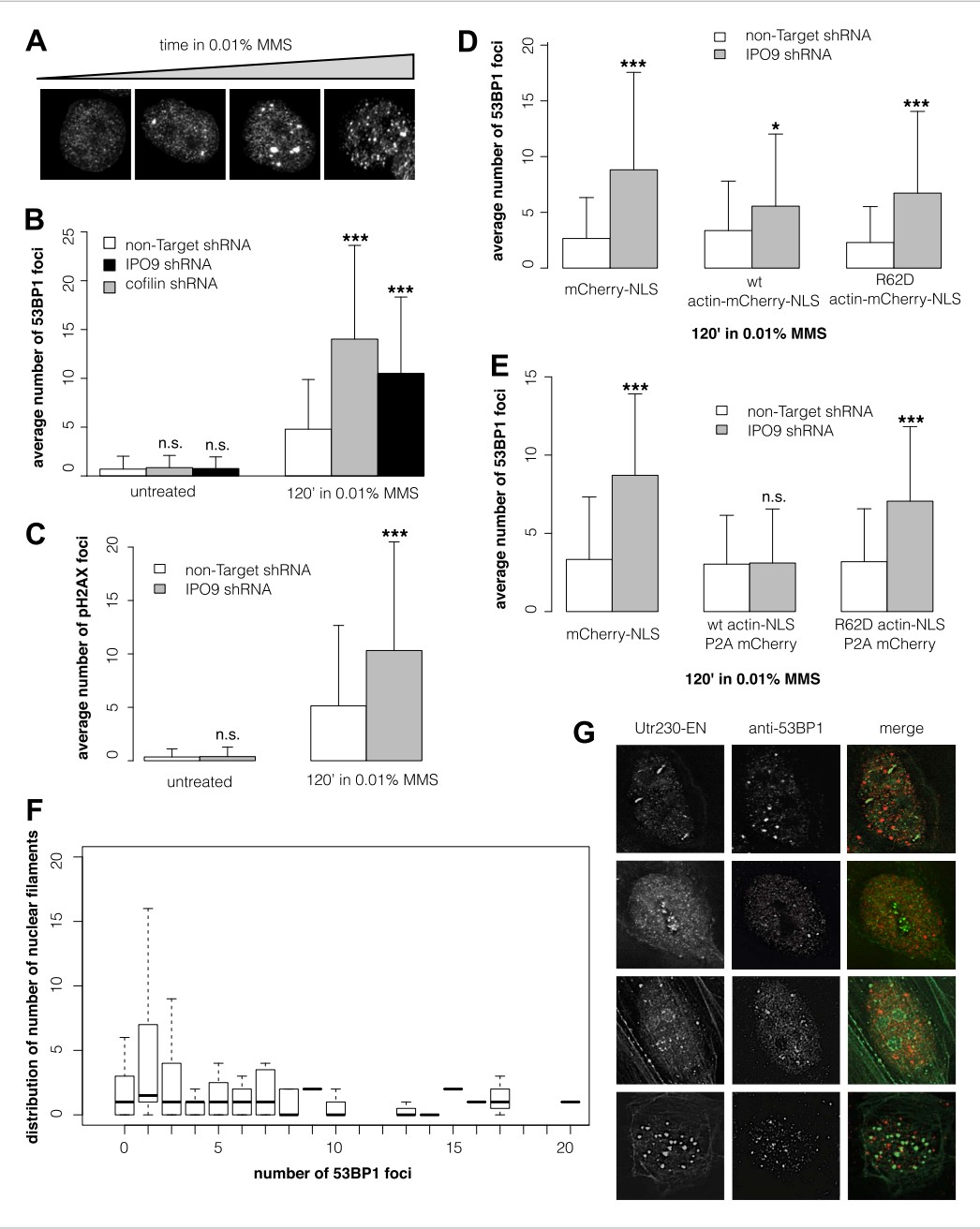

**Figure 3**. Nuclear actin polymerization is required for efficient double-strand break clearance. (**A**) Double-strand break (DSB) sites detected by 53BP1 immunofluorescence after 30′, 60′, 90′, and 120′ incubations in 0.01% MMS. (**B**) Average number of DSB foci detected by 53BP1 immunofluorescence per cell after stable shRNA knockdown of the nuclear actin import factors, importin-9 (IPO9) or cofilin, after 0.01% MMS incubation. N = 125–172 cells per condition. (**C**) Average number of DSB foci detected by gamma H2AX immunofluorescence per cell after stable shRNA IPO9 knockdown following 0.01% MMS incubation. N = 182–241 cells per condition. (**D**) Partial rescue of control DSB foci levels after IPO9 knockdown by overexpression of wild-type actin-NLS-P2A-mCherry but not non-polymerizing R62D mutant of actin-NLS-P2A-mCherry. N = 118–150 cells per condition. (**E**) Full rescue of non-Target control DSB foci levels after IPO9 knockdown by overexpression of wild-type actin-NLS-P2A-mCherry but not non-polymerizing R62D mutant of actin-NLS-P2A-mCherry. N = 262–291 cells per condition. (**F**) Comparison of the distributions of long (>1 micron) nuclear filaments per cell and 53BP1 foci counts. N = 206 cells.
(**G**) Co-localization assays between Utr230-EN and DSBs after 120′ incubation in 0.01% MMS. Asterisks indicate p-values < 10E-2 (*), 10E-3 (**), or 10E-4 (***) for all panels.

*Figure 3. continued on next page*

*Figure 3. Continued*

The following figure supplement is available for figure 3:

**Figure supplement 1**. Knockdown of exportin-6 or IPO9 modulates filament formation after DNA damage.

generated by MMS treatment. Loss of IPO9, however, significantly increased the number of DSBs per cell, judged using antibodies against 53BP1 (*Figure 3B*, *Figure 3—figure supplement 1D*) or pH2AX (*Figure 3C*). We also observed an increase in the number of DSB foci when we depleted nuclear actin by knocking down its nuclear import co-factor, cofilin (*Figure 3B*).

To verify that IPO9 knockdown affects the number of DSB foci by decreasing the concentration of actin in the nucleus rather than by some other, off-target effect, we attempted to rescue the phenotype by expressing actin targeted to the nucleus by an IPO9-independent mechanism. In these experiments, we expressed either: an mCherry-3×NLS control construct; wild-type actin fused to mCherry-3×NLS; or a non-polymerizing R62D actin mutant fused to mCherry-3×NLS. Overexpressing wild-type actin-mCherry-NLS—but not the R62D mutant actin construct—decreased the number of DSB foci but did not completely rescue the loss of IPO9 (*Figure 3D*). We figured that the partial rescue could be due to the fact that fluorescent actin fusions cannot be incorporated into some actin filaments, particularly those generated by formin-family nucleation factors (*Chen et al., 2012*). We, therefore, modified our rescue constructs by inserting a 2A-peptide site to produce bicistronic expression of both a 3×NLS-tagged actin and an mCherry expression reporter (*Kim et al., 2011*; *Huss and Lansford, 2014*). Overexpression of this non-fluorescent, wild-type actin-NLS, but not the R62D mutant, resulted in a complete rescue of DSB foci counts (*Figure 3E*). These results indicate that nuclear actin polymerization is required for efficient DSB clearance and suggested that nuclear actin filaments may be generated by a formin-family actin nucleator.

Finally, we asked whether the number of nuclear actin filaments correlates with the severity of DNA damage by comparing nuclear filaments and 53BP1 foci in the same cells. Surprisingly, we found no correlation between the numbers of nuclear actin filaments and DSBs (*Figure 3F*) and we observed only limited overlap between the 53BP1 and nuclear actin signals for all classes of DNA damage-induced filaments (*Figure 3G*). More detailed analysis of our co-localization data revealed that, out of 198 filaments found in nuclei of cells treated with 0.01% MMS for 2 hr, only 22 (11.1%) overlap or lie adjacent to 53BP1 foci. We cannot, however, rule out the possibility that some fraction of the observed overlap represents a functional interaction. One way to address this possibility would be to simultaneously track DSBs and nuclear actin filaments in live cells, but this would require a more reliable live-cell DSB marker. Given that expression of our fluorescent protein-tagged 53BP1 derivative is correlated with higher foci counts (*Figure 3—figure supplement 1A*), which may include numerous spuriously detected loci, we found this tool to be unsuitable for quantitative live-cell imaging.

## DNA damage-induced filaments do not contain cofilin and are not affected by knockdown of mDia1 or mDia2

We hypothesized that the DNA damage-induced nuclear actin structures we observe might be created by the same mechanisms that produce cofilin–actin rods in response to cell stress (*Fukui, 1978*; *Iida et al., 1992*) or nuclear actin filaments in response to serum stimulation (*Baarlink et al., 2013*). The molecular mechanisms that produce cofilin–actin rods are not well understood, but as their name suggests, they contain the actin-binding protein cofilin (*Munsie et al., 2012*). Due to the poor performance of commercially available antibodies in immunofluorescence, we raised our own polyclonal antibody against human cofilin in rabbit (Covance) capable of labeling nuclear cofilin–actin rods induced by treatment of cells with 10% DMSO. In contrast to the nuclear filaments generated by DMSO, the nuclear actin structures triggered by 0.01% MMS contained no detectable cofilin (*Figure 4A*).

Nuclear actin filaments generated in response to serum stimulation require the activity of formin-family actin nucleators, mDia1 and mDia2 (*Baarlink et al., 2013*). To test whether mDia1 or mDia2 contribute to the production of DNA damage-induced actin filaments, we knocked down their expression using shRNAs. Compared to cells treated with a non-target hairpin, we found no effect of knocking down either mDia1 or mDia2 on the abundance or morphology of nuclear filaments produced by MMS treatment (*Figure 4B,C*; *Figure 4—figure supplement 1*). We conclude that DNA damage-induced filaments represent a biologically and biochemically distinct class of nuclear actin structures.

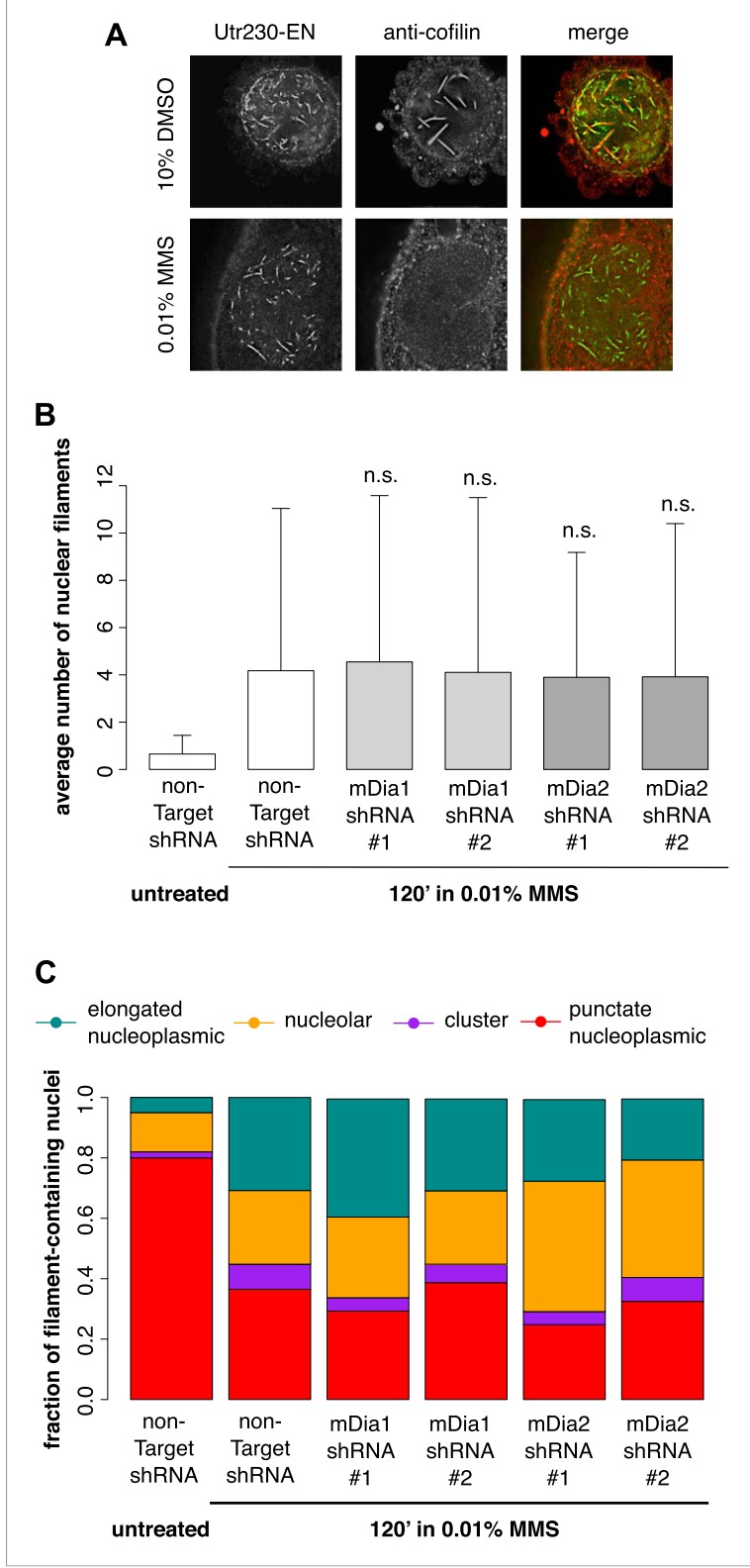

**Figure 4**. DNA damage-induced nuclear actin filaments are distinct from cofilin–actin rods and mDia1/2-generated nuclear filaments. (**A**) Cofilin antibody staining in cells expressing Utr230-EN and treated with either 10% DMSO for 30′ or 0.01% MMS for 120′. (**B**) Average number of nuclear actin filaments per cell after 120′ 0.01% MMS treatment in cells stably expressing mDia1, mDia2, or non-Target control hairpins. N = 115–128 cells per condition.

*Figure 4. continued on next page*

*Figure 4. Continued*

(**C**) Distribution of nuclear actin filament classes after 120′ 0.01% MMS treatment in cells stably expressing mDia1, mDia2, or non-Target control hairpins. N = 115–128 cells per condition.
The following figure supplement is available for figure 4:

**Figure supplement 1**. Validation of mDia1 and mDia2 knockdown by Western blot.

## Formin-2 is required for nuclear actin polymerization in response to DNA damage

To identify candidate regulators of DNA damage-induced nuclear actin assembly, we sifted through a list of mammalian proteins found by *Matsuoka et al. (2007)* to be phosphorylated in response to DNA damage. This list contains many proteins not previously associated with DNA repair, including several known actin regulators. One such protein is the actin nucleator Formin-2 (FMN2). In addition to its phosphorylation, the expression of FMN2 is upregulated in response to both ultraviolet radiation and hypoxia (*Yamada et al., 2013a*). To test whether FMN2 is involved in nuclear actin assembly, we used shRNAs to deplete it from cells expressing the Utr230-EN nuclear actin probe (*Figure 5—figure supplement 1*). When we induce DNA damage with 0.01% MMS, depletion of FMN2 completely inhibits nuclear actin assembly. Both the average number of elongated filaments and the distribution of filament types are nearly identical between MMS-treated, FMN2-knockdown cells and untreated controls expressing a non-target hairpin (*Figure 5A,B*).

To verify the specificity of our hairpin RNA constructs, we attempted to rescue FMN2 knockdown by overexpressing an mCherry-FMN2 variant containing seven silent, hairpin-resistant mutations. Ectopic expression of this mCherry-FMN2 mutant completely restored the number and morphology of the actin structures generated by MMS treatment (*Figure 5C,D*). As with IPO9 knockdown, we found that FMN2 knockdown produces a significant increase in DSB foci observed following MMS treatment. The increase in 53BP1 foci was also rescued by overexpression of hairpin-resistant mCherry-FMN2 (*Figure 5E,F*). Since FMN2 homologs from other vertebrate organisms interact with Spire-family actin nucleators, we knocked down human Spire family members Spire-1 and Spire2 and determined their effect on DNA damage-induced filament assembly (*Figure 6—figure supplement 2*). Knocking down Spire-1 or Spire-2 individually has little effect on DNA damage-induced filament formation (*Figure 6—figure supplement 1*), but knocking down expression of both proteins at the same time produces the same effect as knocking down FMN2, ablating MMS-induced nuclear filament formation (*Figure 6*).

Previous work revealed that human FMN2 translocates from the cytoplasm into the nucleus in response to hypoxia (*Yamada et al., 2013b*). To determine whether DNA damage also induces nuclear translocation, we imaged subcellular distribution of mCherry-FMN2 before and after incubation with 0.01% and 0.05% MMS. We observed that this DNA-damaging agent produces a dose-dependent nuclear accumulation of FMN2 (*Figure 7A,B*; *Figure 7—figure supplement 1*), consistent with a direct role for FMN2 in the assembly of actin filaments in the nucleus following DNA damage. To better understand the nuclear accumulation of FMN2, we searched for a potential nuclear localization sequence (NLS) in human FMN2 using the online bioinformatics tool, cNLS-Mapper (*Kosugi et al., 2009*). We identified two putative NLS regions near the N-terminus of FMN2, beginning at amino acids 6 and 411 of the NCBI reference sequence, accession number NP_064450.3 (*Figure 7C*). To determine whether these regions are functional NLSs, we fused each to an EGFP reporter and expressed it in HeLa cells. Both sequences were sufficient to drive accumulation of EGFP in the nucleus, with stronger nuclear targeting observed for EGFP-NLS2 (*Figure 7D,E*). Mutation of NLS2 at three positively charged residues, K414A/R415A/R416A, efficiently blocks nuclear accumulation of mCherry-FMN2 in response to DNA damage (*Figure 7F*).

## Nucleolar size decreases during the DNA damage response

To determine whether nucleolus-associated actin filaments are involved in regulating a DNA damage-specific change in nucleolar morphology, we labeled nucleoli using fibrillarin and measured (i) the

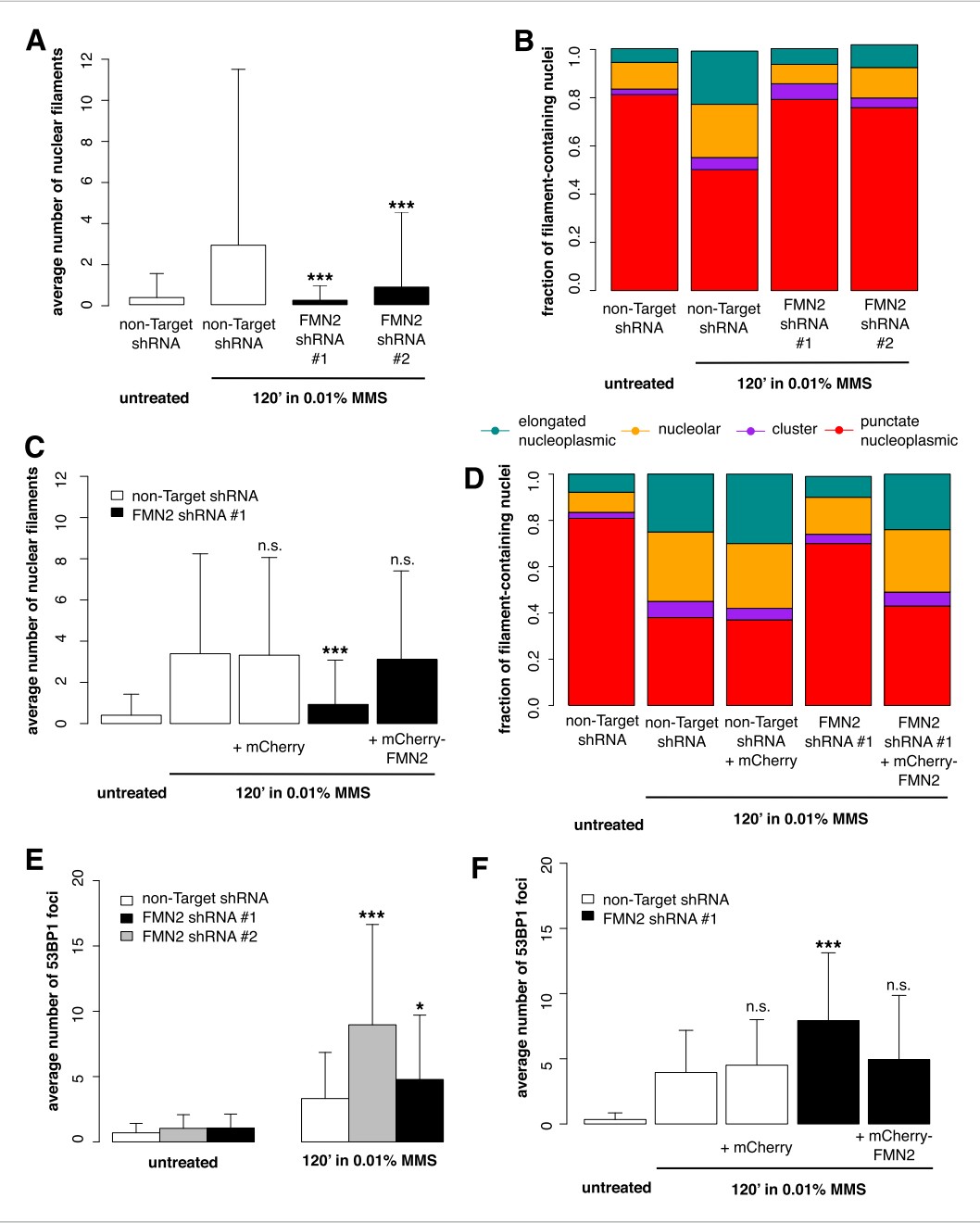

**Figure 5**. Formin-2 is required for DNA damage-induced nuclear actin polymerization. (**A**) Average number of nuclear actin filaments per cell after 120′ 0.01% MMS treatment in cells stably expressing Formin-2 (FMN2) or non-Target control hairpins. N = 117–151 cells per condition. (**B**) Distribution of nuclear actin filament classes after 120′ 0.01% MMS treatment in cells stably expressing FMN2 or non-Target control hairpins. N = 117–151 cells per condition. (**C**) Rescue of FMN2 hairpin in (**A**) by overexpression of hairpin-resistant mutant of mCherry-FMN2. N = 105–127 cells per condition. (**D**) Rescue of FMN2 hairpin in (**B**) by overexpression of hairpin-resistant mutant of mCherry-FMN2. N = 105–127 cells per condition. (**E**) Knockdown of FMN2 via stably selected hairpins increases the number of DSB foci after 0.01% MMS incubation. N = 238–323 cells per condition. (**F**) Rescue of control DSB foci levels after knockdown of FMN2 by overexpression of hairpin-resistant mutant of mCherry-FMN2. N = 229–264 cells per condition. Asterisks indicate p-values < 10E-2 (*) or 10E-4 (***) for all panels.

The following figure supplement is available for figure 5:

**Figure supplement 1**. Validation of FMN2 knockdown by Western blot.

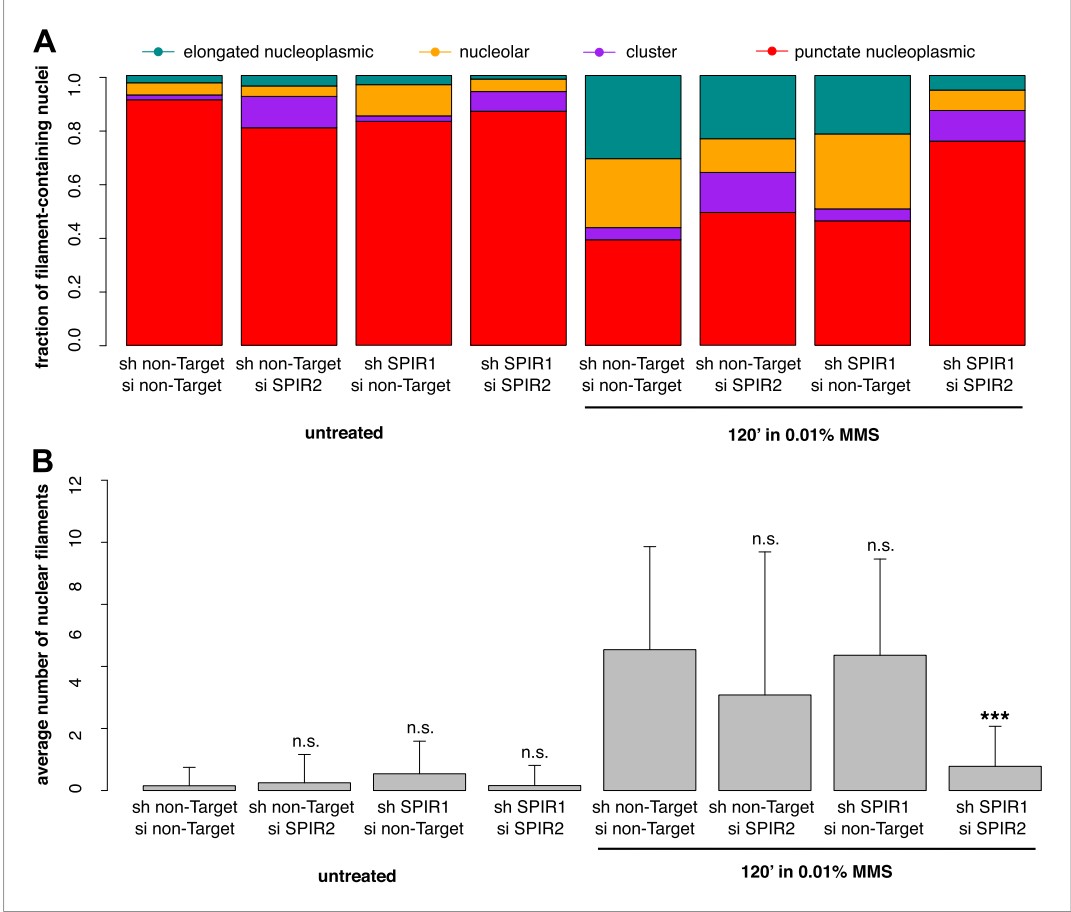

**Figure 6**. Composite knockdown of Spire-1 and Spire-2 inhibits DNA damage-induced nuclear actin assembly. (**A**) Distribution of nuclear actin filament classes after 120′ 0.01% MMS treatment in cells both stably expressing either Spire-1 (SPIR1) or non-Target control hairpins and transiently transfected with either Spire2 (SPIR2) or non-Target control siRNA. N = 111–185 cells per condition. (**B**) Average number of nuclear actin filaments per cell for knockdown conditions described in (**A**). Asterisks indicate p-values < 10E-3 (**) or 10E-4 (***).

The following figure supplements are available for figure 6:

**Figure supplement 1**. Individual knockdown of either Spire-1 or Spire2 does not affect nuclear actin filament polymerization in response to DNA damage.

**Figure supplement 2**. Validation of Spire-1 and Spire-2 knockdowns by Western blot.

number of nucleoli per cell and (ii) nucleolar area after MMS treatment (*Figure 8*). We performed these experiments in both control cells expressing a non-target hairpin and an IPO9 shRNA knockdown line. Surprisingly, while the average number of nucleoli is constant, induction of DNA damage decreases nucleolar size by an average of ~20%. Equivalent decreases in nucleolar area were observed in both control and IPO9 knockdown cells, however, indicating that the nucleolar area reduction is nuclear actin-independent.

## Actin regulates nuclear oxidation after acute DNA damage

In some cases, DNA damage has been shown to result in changes in the oxidative state of the nucleus, and it has been recently shown that oxidative stress can activate a subset of DNA-repair signaling cascades independently of DNA damage (*Guo et al., 2010*; *Ditch and Paull, 2012*). Oxidative pathways also regulate previously studied cases of stress-induced nuclear actin polymerization

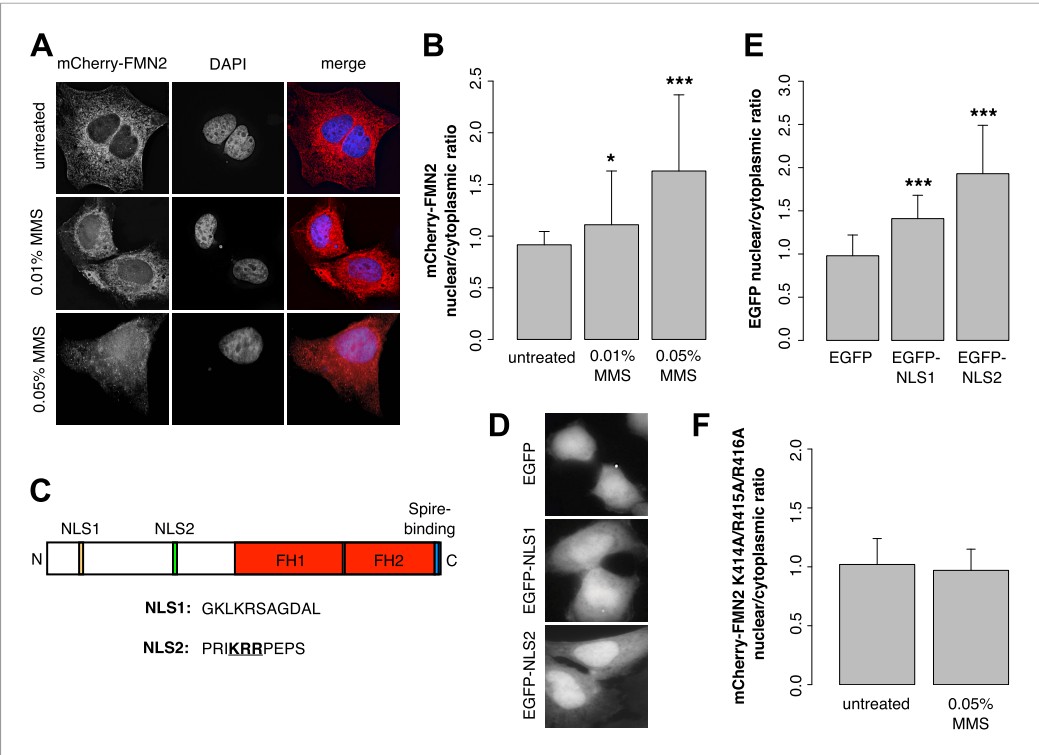

**Figure 7**. Formin-2 accumulates in the nucleus in response to DNA damage. (**A**) Localization of transiently expressed mCherry-FMN2 in untreated cells and after 120′ incubation in 0.01% or 0.05% MMS. (**B**) Average ratio of nuclear vs cytoplasmic integrated fluorescence intensity of mCherry-FMN2 in untreated cells and after 0.01% and 0.05% MMS. N = 71–94 cells per condition. (**C**) Domain organization of human FMN2 including two putative nuclear localization sequence (NLS) sites identified using cNLS Mapper (*Kosugi et al., 2009*). (**D**) Localization of EGFP fused to each putative NLS. (**E**) Nucleocytoplasmic ratio of EGFP fused to each putative NLS. N = 140–165 cells.
(**F**) Nucleocytoplasmic ratio of the K414A/R415A/R416A NLS2 mutant mCherry-FMN2 before and after 0.05% MMS treatment. N = 108–115 cells. Asterisks indicate p-values < 10E-3 (**) or 10E-4 (***) for all panels.

The following figure supplement is available for figure 7:

**Figure supplement 1**. Gallery of mCherry-FMN2 localization before and after MMS treatment.

---

(*Pfannstiel et al., 2001*; *Bernstein et al., 2012*; *Lundquist et al., 2014*). To assess whether oxidative signaling is involved in regulation of nuclear actin filaments induced by MMS, we generated a construct containing the redox-sensing fluorophore roGFP2 fused with 3×NLS (roGFP2-NLS) (*Lohman and Remington, 2008*). roGFP2 is variant of Green Fluorescent Protein (GFP) engineered to introduce 2 cysteines to the interior of the GFP beta barrel, and it has been extensively used to measure oxidation changes in live cells (*Merksamer et al., 2008*; *Al-Mehdi et al., 2012*). The roGFP2 excitation spectrum contains 2 peaks, at 488 nm and at 405 nm. In the oxidized form of roGFP2, excitation at 405 nm is increased and excitation at 488 nm is decreased, relative to the reduced form. Thus, the ratio of emission intensities following excitation of roGFP2 at these wavelengths under different conditions can be used to measure the relative oxidation state.

We used roGFP2-NLS to measure nuclear oxidation levels and found that treatment with 0.01% MMS generates an oxidative burst in the nucleus (*Figure 9A*). Depletion of nuclear actin by IPO9 knockdown has no effect at this MMS dosage (*Figure 9B*). Curiously, however, when the MMS treatment was increased to 0.05%, IPO9 knockdown abolished nuclear oxidation; we observed the same result following knockdown of FMN2 (*Figure 9C–E*). We measured the distribution of classes of nuclear actin filaments at MMS doses ranging from 0.01% to 0.05%, and found that increased MMS levels correlate with increases in the fraction of cells containing clustered bodies of nuclear filaments (*Figure 9F*). These results suggest that clustered filaments may participate in regulating the nuclear redox state.

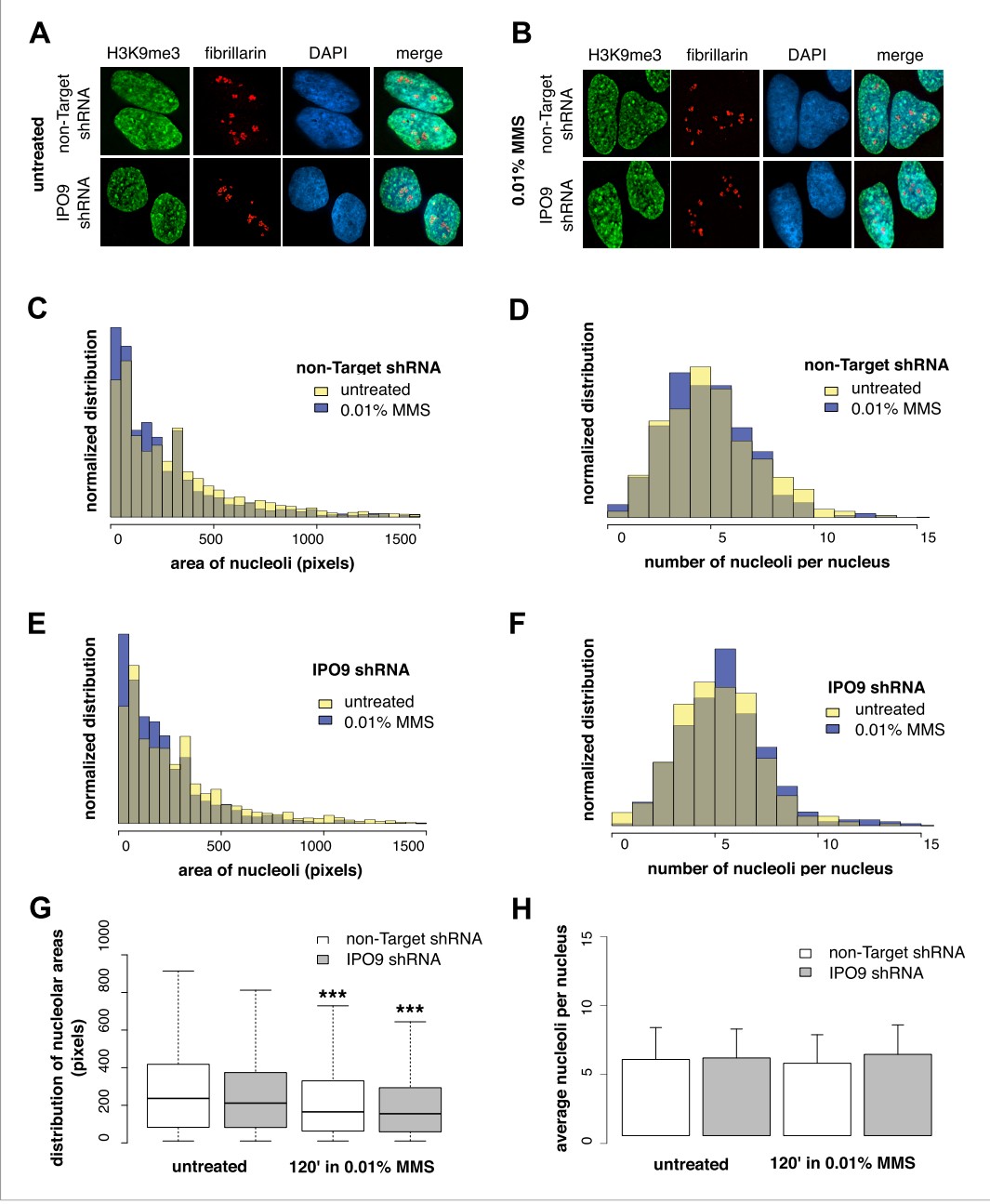

**Figure 8**. Nucleolar size decreases after DNA damage through a nuclear actin-independent pathway. (**A**) Localization by immunofluorescence of heterochromatin marker H3K9me3 and nucleolar marker fibrillarin in untreated cells stably expressing non-Target and IPO9 hairpins. (**B**) Repeat of (**A**) following 0.01% MMS incubation for 120′. (**C**) Distribution of nucleolar areas (in pixels) before (N = 2313 nucleoli) and after 0.01% MMS incubation (N = 2299 nucleoli) in IPO9 knockdown cells. (**D**) Distribution of nucleolar areas (in pixels) before (N = 2094 nucleoli) and after 0.01% MMS incubation (N = 2180 nucleoli) in control cells. (**E**) Distribution of average numbers of nucleoli per cell before (N = 406 cells) and after 0.01% MMS incubation (N = 386 cells) in IPO9 knockdown cells. (**F**) Distribution of average numbers of nucleoli per cell before (N = 375 cells) and after 0.01% MMS incubation (N = 411 cells) in control cells. (**G**) Box plots comparing distributions shown in (**C**, **D**). Asterisks indicate p-values < 10E-4 (***). (**H**) Bar plots comparing averages from distributions shown in (**E**, **F**).

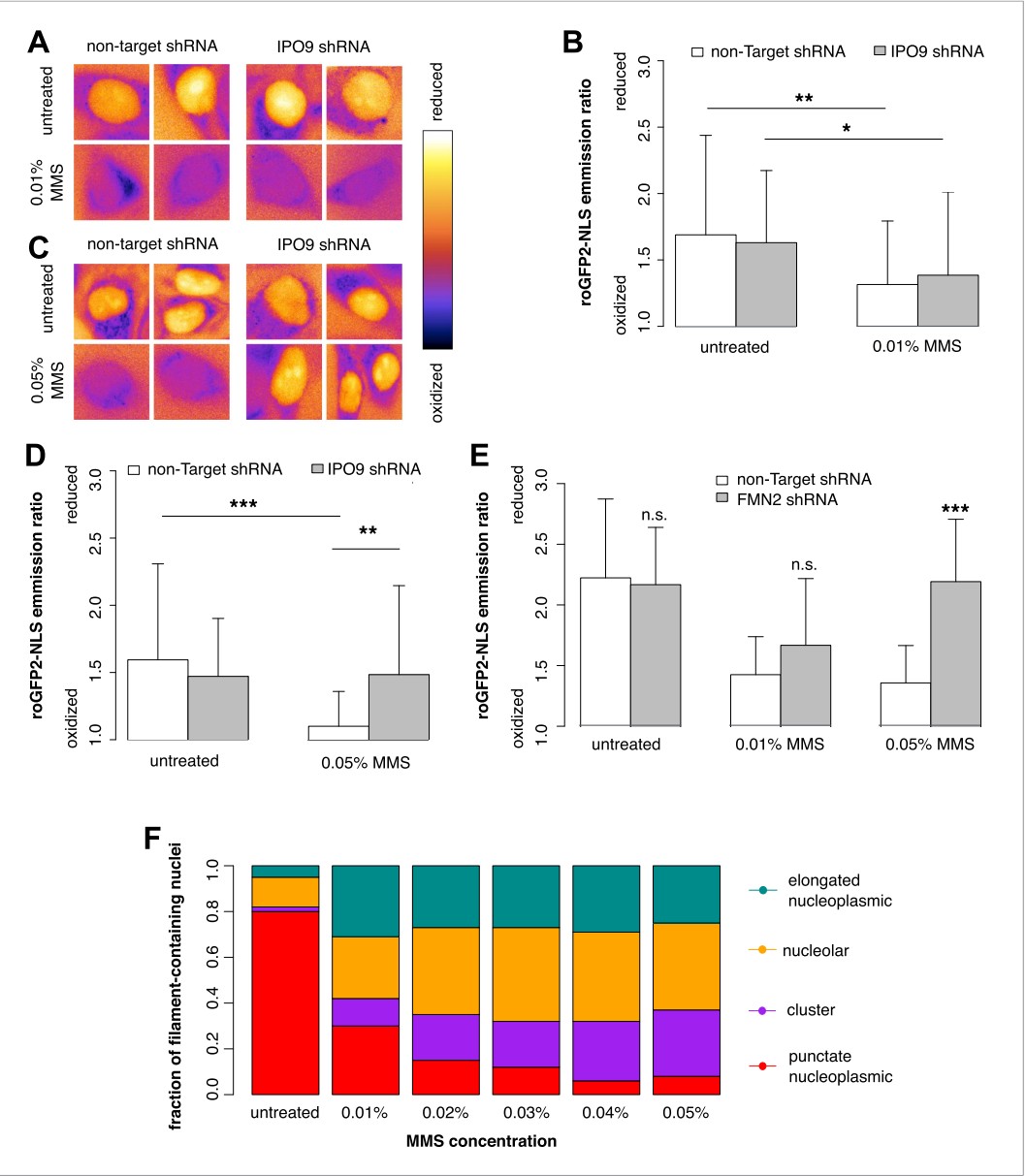

**Figure 9**. Nuclear actin is required for regulation of nuclear oxidation after acute DNA damage. (**A**) False-colored images of roGFP2-NLS signal (ratio of emission intensities after 488-nm and 405-nm excitation) after 120′ in 0.01% MMS in non-Target control and IPO9 knockdown cell lines. (**B**) Average roGFP2-NLS signal integrated across the nuclear area in non-Target control or IPO9 hairpin lines in untreated and 0.01% MMS-treated cells. N = 108–144 cells per condition. (**C**) False-colored images of roGFP2-NLS signal (ratio of emission intensities after 488-nm and 405-nm excitation) after 120′ in 0.05% MMS in non-Target control and IPO9 knockdown cell lines. (**D**) Average roGFP2-NLS signal in nuclei of non-Target control or IPO9 hairpin lines in untreated and 0.05% MMS-treated cells. N = 106–124 cells per condition. (**E**) Average roGFP2-NLS signal in the nuclei of cells stably expressing non-Target or FMN2 shRNA in untreated cells and cells treated with 0.01% or 0.05% MMS for 120′. N = 109–162 cells. (**F**) Distribution of nuclear actin filament classes after incubation in MMS concentrations from 0.01% to 0.05% for 120′. N = 95–159 cells per condition. Asterisks indicate p-values < 10E-3 (**) or 10E-4 (***) for all panels.

## Discussion

Biochemical, cell biological, and genomic studies have revealed many ancient and important connections between actin family proteins and DNA (*Belin and Mullins, 2013*), but we still know little about the functions of actin inside the nucleus. For example, many chromatin-remodeling complexes

contain monomeric actin or Arps as subunits, but the contribution these subunits make to chromatin remodeling remains poorly understood. In addition, cellular stresses such as ATP and nutrient deprivation, heat shock, and various chemical toxins induce the assembly of cytoplasmic actin bundles as well as cofilin–actin rods in the nucleus. Although they are thought to sense reactive oxygen species (*Bernstein et al., 2012*), the precise function of nuclear cofilin–actin rods remains unclear. One of the most convincing functions proposed for nuclear actin is regulation of the actin-binding transcription factor MRTF. In this example, serum stimulation of quiescent cells induces assembly of actin filaments in both the cytoplasm and the nucleus (*Baarlink et al., 2013*). Assembly of filaments in the cytoplasm drives MRTF into the nucleus, while the assembly of filaments inside the nucleus depletes monomeric actin from this compartment and relieves its inhibitory effect on MRTF (*Vartiainen et al., 2007*). This mechanism relies on the nucleation activity of Diaphanous-family formins, mDia1 and mDia2, and highlights the existence of a stable pool of polymerizable actin monomers inside the nucleus.

We have uncovered another, previously unknown signaling pathway that generates actin filaments inside the nucleus, one that relies on the Cappuccino-family formin, FMN2, and two Spire-family actin regulators, Spire-1 and Spire-2. In response to DNA damage, these nucleation factors create a variety of actin structures in the nucleus, including long nucleoplasmic filaments, nucleolus-associated filaments, and amorphous clusters. These may represent functionally distinct species or possibly different time points in the evolution of a single structure. Regardless, as judged by the number of 53BP1 foci, nuclear actin assembly promotes clearance of double-strand DNA breaks. In contrast to the role of nuclear actin in MRTF function, however, filament formation appears to be more important than monomer depletion in the response to DNA damage. A functional role specific to filaments is supported by the fact that removing *all* actin (monomer and polymer) from the nucleus has the same effect on MMS-induced 53BP1 foci as simply inhibiting filament formation by depleting the nucleation factor, FMN2. If monomer depletion were the primary function of DNA damage-induced filaments, then IPO9 knockdown would not have the same effect on DNA repair as filament inhibition.

Little work has been done on human FMN2, but mouse and *Drosophila* FMN2 homologs are relatively well studied. The nucleation activity of the *Drosophila* FMN2 homolog, Cappuccino, is regulated by intramolecular contacts between a C-terminal region near the actin-nucleating domain and an N-terminal sequence (*Bor et al., 2012*). This autoinhibitory interaction is structurally distinct from the DID/DAD interaction that regulates activity of the Diaphanous-family formins. We have identified two nuclear-localization sites in the N-terminus of human FMN2, which also contains two previously discovered DNA damage-induced phosphorylation sites. If human FMN2 is also autoinhibited by an interaction between N- and C-terminal regions, it is possible that DNA damage-induced phosphorylation promotes both nuclear translocation and actin nucleation.

Nuclear actin filaments could potentiate the DNA damage response by several mechanisms: (1) altering mobility or organization of chromatin; (2) recruiting or delivering repair factors to sites of damage; or (3) binding and sequestering nuclear factors that would otherwise inhibit DNA repair. The third possibility is consistent with the minimal co-localization we observe between nuclear actin filaments and 53BP1 foci. Recent studies, however, have also demonstrated that DNA damage alters the mobility of DNA loci, perhaps facilitating their interaction with DNA repair proteins that are expressed at extremely low concentrations (*Dimitrova et al., 2008*; *Miné-Hattab and Rothstein, 2013*). One attractive possibility is that myosin motors in the nucleus may drive clustering of repair components or DSB sites to facilitate homologous recombination. This would be analogous to the role of FMN2/Spire-generated actin filaments in generating contractile structures that link cargo vesicles in mouse oocytes.

The DNA damage-induced recruitment of actin filaments to nucleoli is somewhat mysterious. We initially hypothesized that the nucleolar filaments may regulate some DNA damage-specific change in nucleolar morphology, analogous to the role of actin filaments in germinal vesicles of *Xenopus* oocytes. Although we observed a 20% decrease in average nucleolar area after MMS treatment, depleting nuclear actin by knocking down IPO9 had no measurable effect on this phenomenon. Given that nucleoli are the sites of rRNA transcription, we speculate that nuclear actin filaments may play a role in rRNA production or export, perhaps promoting stress-induced changes in translation. Clarifying the potential nucleolar functions of actin will require the identification of more nucleolus-associated actin-binding proteins.

In addition to filamentous structures, DNA damage also induces formation of amorphous actin clusters in the nucleus. The number and size of these clusters increases with increasing doses of genotoxic agents, suggesting that they may be related to pre-apoptotic signaling. Consistent with this idea, we found that blocking nuclear filament assembly inhibits nuclear oxidation induced by high

concentrations of MMS. Several prior studies suggested that actin acts as a sensor of cellular oxidation levels. For example, in the yeast *Saccharomyces cerevisiae*, cellular oxidation triggers formation of amorphous 'actin bodies' generated by intermolecular disulfide bonds (*Farah and Amberg, 2007*, *2011*). Similar sensing of nuclear oxidation has also been proposed for nuclear cofilin–actin rods, which contain disulfide-linked cofilin oligomers (*Bernstein et al., 2012*). While more work will be required to determine whether DNA damage-induced nuclear filament clusters are also oxidation sensors, we demonstrate clearly that actin can alter the nuclear redox state. Given that all previously observed cases of nuclear actin assembly are connected to oxidation, we suggest that nuclear actin may be an important mediator of redox signaling.

Together with other recent studies, our work reveals that multiple cellular signals can trigger assembly of a variety of actin structures in the nucleus, each with a unique morphology, function, and regulatory mechanism. One common theme, however, appears to be a link between the state of actin in the nucleus and the state of actin in the cytoplasm. For example, both serum stimulation (*Vartiainen et al., 2007*; *Baarlink et al., 2013*) and DNA damage (*Zuchero et al., 2012*; and unpublished observations) induce a dramatic increase in cytoplasmic actin polymerization. In both cases, this cytoplasmic actin polymerization correlates with nuclear accumulation of formin-family actin nucleators. We, therefore, speculate that changes in the actin monomer/polymer ratio in the cytoplasm may facilitate nuclear accumulation of actin regulators, coordinating the cytoplasmic and nuclear responses to stimuli.

## Materials and methods

### Molecular biology

We used pEGFP-C1 or pmCherry-C1 (Clontech, Mountain View, CA, United States) as the host vector for all EGFP fusions, with N-terminal EGFP fusions inserted into the unique AgeI and NheI sites. Lifeact and all NLS sequences were cloned directly using annealing primers purchased from Elim Biopharmaceuticals. F-tractin and 3×NLS-peptide 2A inserts were synthesized and inserted into backbone vectors using custom cloning services from GenScript. Human actin was subcloned from cDNA purchased from GE Dharmacon (formerly Open Biosystems). 53BP1 fragment was subcloned from Addgene plasmid #19835 (*Dimitrova et al., 2008*) and was provided by Beth Cimini. mCherry-FMN2 was provided by Sonia Rocha and includes the following N-terminal insertion in comparison to the NCBI human FMN2 reference sequence (accession number NP_064450.3): AGATCTCATTCGA TTCGCACGGTGGAGATTAAAGTCCCCGAGATAGAGGAAACGTTTTTCGCGCCCAGGTTCAGCGAG GAGCCGCGCGGGGGGCAGAGGGGGCGGCGGCGGCGGGCGGGGAGCCAGGCCCGAGCTGCGTT CTGCGCAGCCATTGGTGGGCGCCGCACTCTGCACTGAGCATGTTCGCGCCCCGCCGGCCCCTAG CCGCAGCCGCAGCCGCAGCGACGGCAGCCACGGGAGCCGCCGCGCATTATGCAAAGCGGCGG CAGATGCGAGCGGGGCCAGCCGGGCGCGCGTCGGCCTCCCCTCCCAGCGGCTCCCCCCGCCGC CGCCTGACTCTCCCGGGAGACTCCCTAGGCCCGGGATTGCACC. roGFP2 was a gift from Philip Merksamer and Ferroz Papa. Lentiviral packaging vectors were provided by John James. Mutagenesis of mCherry-FMN2 was performed using a Q5 polymerase site-directed mutagenesis (SDM) kit according to the manufacturer's protocol, modified to include the Q5 High GC Enhancer and an extension temperature of 75°C (New England Biolabs, NEB). Primer sequences for SDM were generated using the companion NEB primer design tool. All new constructs generated for this study are being made available on Addgene.

### Cell culture

HeLa cells and HEK293T lentivirus-generating cells (ATCC) were cultured in Dulbecco's modified Eagle's medium supplemented with 10% Fetal Bovine Serum (FBS, Atlas Biologicals, Fort Collins, CO, United States), 2 mM L-glutamine and penicillin–streptomycin (UCSF Cell Culture facility) at 37°C with 5% $CO_2$. For transient transfection of non-knockdown constructs in HeLa cells, cells were transfected using Lipofectamine LTX (Life Technologies) according to the manufacturer's protocol. All transient transfections were performed 24–72 hr prior to data collection. Stable cell lines of Utr230-EGFP-NLS were generated as described in *Belin et al. (2013)*. For knockdown constructs, lentiviral packaging vectors pMDG.2 and pCMVΔ8.91 were co-transfected with shRNA vectors into HEK293T cells using GeneJuice (EMD Millipore) as described in *James and Vale, 2012*. Lentiviral particles were harvested 2 and 3 days following transfection, sterile-filtered using 0.2-μm Steriflip filter units (EMD Millipore) and either used immediately or stored long-term at −80°C.

## Knockdown reagents

Mock siRNA and Silencer Select siRNAs directed against human IPO9, XPO6, and Spire2 were purchased from Life Technologies. Transient reverse transfection was performed using Lipofectamine RNAiMAX (Life Technologies) according to the manufacturer's protocol. At 8–24 hr following transient transfection, the cell medium was replaced. Cells were split into flasks and/or fibronectin coverslips 3 days after transfection and were either fixed for imaging or lysed for Western blotting at 5 days following transfection. Non-Target shRNA and Mission shRNAs directed against mDia1, mDia2, cofilin, FMN2, Spire-1, and IPO9 were purchased as bacterial glycerol stocks from Sigma, and lentiviral particles were generated as described above. HeLa cells were transduced with lentiviral particles and split after 8–16 hr into media supplemented with 0.3 µg/ml puromycin (Thermo Fisher Scientific) for stable selection. All knockdown constructs were pre-validated by qRT-PCR using the SuperScript III Platinum SYBR Green One-Step qRT-PCR kit (Life-Technologies) and run on a Stratagene Mx3005P RT-PCR thermocycler (UCSF Center for Advanced Technology). RNA was insolated using the TRIzol Plus RNA Purification Kit according to the manufacturer's instructions (Life Technologies). qRT-PCR primers were designed using Primer BLAST (NIH). Knockdowns verified by qRT-PCR were subsequently validated by Western blotting (performed as described in *Belin et al., 2013* but modified to include 1× Detector Block [KPL/Sera Care Life Sciences] as the primary antibody diluent).

## Antibodies

Our cofilin antibody was generated in rabbit by Covance from purified human cofilin; sera from inoculated rabbits rather than purified antibody was used in immunofluorescence assays. Commercial primary antibodies used in this study follow: anti-tubulin (Sigma, Cat. No. T29026), anti-fibrillarin (Abcam, Cat. No. ab4566), H3K9me3 (Abcam, Cat. No. ab8898), anti-53BP1 (Novus Biologicals, Cat. No. NB100-305), anti-H2AX pS139 (Millipore, Cat. No. 05–636, Clone JBW301), IPO9 (Abcam, Cat. No. ab52605), mDia2 (ECM Biosciences, Cat. No. DP3491), mDia1 (BD Biosciences, Cat. No. 610848), FMN2 (Abnova, Cat. No. H00056776-A01), Spire-1 (Aviva Systems Biology, Cat. No. OAAB12896), Spire2 (ThermoFisher—Pierce, Cat. No. PA5-24099).

## DNA damage assays

For telomere uncapping assays, MT-hTer-47A shRNA cell lines were was generated as described in *Stohr and Blackburn (2008)* and cells were fixed for imaging one week after initial infection. For global DNA damage induction via UV, cells were washed three times in calcium- and magnesium-free Phosphate-Buffered Saline (PBS). The PBS was removed and cells were exposed to 50 J/m-s UV in a hybridization oven. After exposure, cells were washed once and complete medium was added. UV-treated cells were then incubated at 37°C with 5% $CO_2$ for 6 hr and then fixed for imaging. For neocarzinostatin (NCS; Sigma) treatment, NCS was added to complete medium to a final concentration of 50 pg/ml. Cells were washed in PBS and NCS-supplemented media was added; cells were then incubated at 37°C with 5% $CO_2$ for 6 hr and fixed for imaging. For methyl methanosulfonate (MMS; Sigma) treatment, MMS was added to complete medium at a final concentration of either 0.01% or 0.05% vol/vol. Cells were washed in PBS and MMS-supplemented media was added; cells were then incubated at 37°C with 5% $CO_2$ for 2 hr or as indicated before fixation for imaging.

## Immunofluorescence and staining

Cells were passaged onto glass coverslips coated with 10 µg/ml fibronectin (Invitrogen) and cultured overnight to 30–60% confluence. Coverslips were fixed at room temperature for 15–30 min in 4% paraformaldehyde (prepared from 16% solution from Electron Microscopy Sciences) in PBS (UCSF Cell Culture Facility). Cells were permeabilized in 0.1% Triton-X-100 (Sigma) in PBS for 3–5 min and blocked in 1× blocking buffer (abcam)/PBS at room temperature for 60 min. Primary antibody incubations were performed for 60 min at room temperature or overnight (anti-53BP1) in 1× blocking buffer (abcam)/PBS. Cells were incubated in Alexa Fluor-labeled secondary antibodies (Invitrogen) in 1X blocking buffer (abcam)/PBS for 30 min at room temperature and mounted on slides Fluoromount G with DAPI (Affymetrix eBioscience). For phalloidin staining, cells were fixed as above and stained for 15 min in 0.1% Triton/PBS with 100 µM Alexa Fluor 568-phalloidin (Invitrogen).

## Microscopy and image analysis

Images were acquired using a DeltaVision RT system (Applied Precision) with a Photometrics CoolSnapHQ camera using a 100× 1.40 NA UPlanSApo objective (Olympus). Images were processed for contrast enhancement and background noise reduction using ImageJ (National Institutes of Health). All image analysis was performed using custom ImageJ macros (see *Source code 1*). All figures and statistical analyses were generated in R. Reported p-values are the results of a two-tailed t-test.

## Acknowledgements

This work was primarily supported by grants from the National Institutes of Health to RDM (GM061010 and GM079556) and funding from the Howard Hughes Medical Institute. Additional support was provided by a National Science Foundation Predoctoral Fellowship (BJB) and a National Institutes of Health Ruth L Kirschstein Predoctoral Fellowship (BJB). Some early experiments for this project were conducted at the Marine Biological Laboratory in Woods Hole, MA. We are grateful to Elizabeth Blackburn and Beth Cimini for use of their DeltaVision microscope and a great deal of technical assistance, including generation of the 53BP1-mTag-BFP2 construct, MT-hTer-47A HeLa RNAi line, and DSB antibody selection and staining conditions. Eric Chow of the UCSF Center for Advanced Technology assisted in providing technical support for design and implementation of qRT-PCR experiments. We thank Sonia Rocha of the University of Dundee, Phil Merksamer from the Ferroz Papa lab, and John James formerly of the Ron Vale lab at UCSF for providing multiple constructs used in this study, as noted in the Methods section. Noah Ollikainen provided helpful comments on the manuscript and advice on figure preparation ad statistical analysis in R. Thank you finally to Jenny Hsiao for gifting a generous quantity of purified human cofilin for generating a cofilin antibody.

## Additional information

### Funding

| Funder | Grant reference | Author |
|---|---|---|
| Howard Hughes Medical Institute | | Brittany J Belin, Terri Lee, R Dyche Mullins |
| National Institutes of Health | GM061010, GM079556, 5F31AG39147-2 | Brittany J Belin, Terri Lee, R Dyche Mullins |
| National Science Foundation | | Brittany J Belin |

The funders had no role in study design, data collection and interpretation, or the decision to submit the work for publication.

### Author contributions

BJB, Analysis and interpretation of data; TL, Preparing and editing the manuscript and figures; RDM, Conception and design, Acquisition of data, Analysis and interpretation of data, Drafting or revising the article

## Additional files

### Supplementary file

• Source code 1. Custom ImageJ macros.

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
