## [Decision Letter]

Thank you for submitting your work entitled “Formin-2 and Spire-1/2 create nuclear actin filaments that promote efficient DNA repair during the DNA damage response” for peer review at *eLife*. Your submission has been favorably evaluated by Randy Schekman (Senior Editor) and three peer reviewers, one of whom, Pekka Lappalainen, is a member of our Board of Reviewing Editors, and another is Robert Grosse.

The reviewers have discussed their reviews with one another and the Reviewing Editor has drafted this decision to help you prepare a revised submission.

The three reviewers found the findings presented in the manuscript important and of significant interest, both in the cytoskeleton and chromatin fields. Although a majority of the experiments appear to be of good technical quality, a few additional experiments/data analyses would significantly strengthen the study. Furthermore, as described below, there are several additional points that should be addressed by revising the manuscript text/figures.

1) The data in Figure 1 should be carefully quantified to confirm the conclusions presented in the text (each of the treatments were claimed to trigger nuclear actin filament assembly that was indistinguishable from that produced by telomere uncapping).

2) The data presented in Figure 2 are not particularly convincing. If possible, the authors should provide better quality images or alternatively perform more sophisticated data analysis including, for example, quantification of the average lengths of the nuclear filaments/bundles visualized by phalloidin alone or by the other probes.

3) The authors should also more precisely discuss why only 15-20 % of the cells have detectable nuclear actin filaments following MSS treatment.

4) The possible role of FMN2-assemled nuclear actin filaments in regulation of nuclear oxidation following DNA damage should be tested to better link the data presented in Figure 9 to the study. This could be performed by FMN2 RNAi studies, which would be preferentially combined with rescue experiments using wild-type FMN2 and the mutant FMN2 (K414A/R415A/R416A), which does not accumulate to the nucleus.

---

## [Author Response]

*The three reviewers found the findings presented in the manuscript important and of significant interest both in the cytoskeleton and chromatin fields. Although a majority of the experiments appear to be of good technical quality, a few additional experiments/data analyses would significantly strengthen the study. Furthermore, as described below, there are several additional points that should be addressed by revising the manuscript text/figures*.

*1) The data in*
Figure 1
*should be carefully quantified to confirm the conclusions presented in the text (each of the treatments were claimed to trigger nuclear actin filament assembly that was indistinguishable from that produced by telomere uncapping)*.

We have rewritten this section of the manuscript to moderate our comparison of the actin structures produced by different DNA-damaging agents. We agree with the reviewers’ sense that the amount of data available for the comparison of all the various genotoxic stresses does not support use of such a strong word as “indistinguishable.”

We did, however, perform an additional experiment to confirm that telomere uncapping (5 and 7 days after treatment with RNAi agent) produces an identical nuclear actin response to DNA damage by 0.01% MMS (for two hours). We analyzed data from HeLa cells +/- telomere capping and +/- 0.01% MMS and quantified: (i) the % of cells with >1 micron filaments, (ii) the distribution of the total number of >1 micron filaments/nucleus, and (iii) the distribution of >1 micron filament lengths for all of these conditions.

This new analysis is shown in revised Figure 1.

*2) The data presented in*
Figure 2
*are not particularly convincing. If possible, the authors should provide better quality images or alternatively perform more sophisticated data analysis including e.g. quantification of the average lengths of the nuclear filaments/bundles visualized by phalloidin alone or by the other probes*.

This figure is not central to our argument and our major conclusions regarding the role of FMN-2/Spire-induced actin assembly and its role in the DNA damage do not rely on these results. We decided, however, to include the data in the manuscript for the sake of completeness. We have also analyzed the lengths of nuclear filaments in 0.01% MMS treated cells when visualized by NLS-Lifeact, phalloidin, and NLS-Utr230. This new analysis is shown in revised Figure 2.

We take some exception to this comment, given that this type of experiment is not normally performed in the field. The difficulties in visualizing nuclear actin filaments with phalloidin are well known and part of the motivation for developing more sophisticated nuclear actin probes. We include these data as evidence that, even though it is extremely challenging, we can still observe DNA damage-induced nuclear actin by phalloidin.

*3) The authors should also more precisely discuss why only 15-20 % of the cells have detectable nuclear actin filaments following MSS treatment*.

To address this point we performed additional analysis on the correlation between the number and morphology of nuclear actin filaments and the number of 53BP1 foci (a proxy for double-strand breaks). This analysis reveals no detectable correlation between DSB counts and number of nuclear actin filaments induced by 0.01% MMS. In the revised manuscript we now speculate on the reasons for this lack of correlation.

The new analysis is shown in revised Figure 3.

*4) The possible role of FMN2-assemled nuclear actin filaments in regulation of nuclear oxidation following DNA damage should be tested to better link the data presented in*
Figure 9
*to the study. This could be performed by FMN2 RNAi studies, which would be preferentially combined with rescue experiments using wild-type FMN2 and the mutant FMN2 (K414A/R415A/R416A)*, *which does not accumulate to the nucleus.*

In response to this comment we performed an additional experiment, knocking down FMN-2 and then quantifying nuclear oxidation in response to MMS treatment. Briefly, we imaged and quantified the roGFP-NLS signal with and without 0.05% MMS treatment in cells that expressed either a control shRNA or an shRNA targeting FMN2.

Consistent with our hypothesis, we find that loss of FMN-2 ablates the nuclear oxidation in response to MMS treatment. We were unable to perform the rescue experiment because our rescue constructs are all fused to mCherry, whose fluorescence emission overlaps that of the rosGFP.

These new data and analysis are shown in revised Figure 9.